# Effects of the Fibrous Root of *Polygonatum cyrtonema* Hua on Growth Performance, Meat Quality, Immunity, Antioxidant Capacity, and Intestinal Morphology of White-Feathered Broilers

**DOI:** 10.3390/antibiotics12111627

**Published:** 2023-11-15

**Authors:** Tianlu Zhang, Dong Zhou, Miaofen Chen, Hui Zou, Qi Tang, Ying Lu, Yajie Zheng

**Affiliations:** 1College of Horticulture, Hunan Agricultural University, Changsha 410128, China; ztl15581233625@163.com (T.Z.); zzbllww@163.com (D.Z.); tangqi@hunau.edu.cn (Q.T.);; 2National Research Center of Engineering Technology for Utilization Ingredients from Botanicals, Changsha 410128, China; 3College of Animal Medicine, Hunan Agricultural University, Changsha 410128, China; chenmiaofan2023@163.com; 4Yipuyuan Huangjing Technology Co., Ltd., Xinhua 417600, China; zuohui168@yahoo.com.cn

**Keywords:** fibrous root of *Polygonatum cyrtonema* Hua, antibiotic alternative, growth performance, immune function, white-feathered broiler

## Abstract

This study was designed to evaluate the effects of different doses of the fibrous roots of *Polygonatum cyrtonema* Hua on the growth performance, slaughter parameters, meat quality, immune function, cytokines, antioxidant capacity, and intestinal morphology of white-feathered broilers. Also, the mechanism to improve immune functions of broilers was explored through network pharmacology and molecular docking technology. A total of 360 AA-white-feathered broilers were randomly divided into six groups (not separated by sex), with six repetitions per group (*n* = 10). The groups were as follows: basal diet (CON group), basal diet supplemented with 300 mg/kg aureomycin (ANT group), basal diet supplemented with 2%, 3%, and 4% fibrous root raw powder (LD, MD, and HD group), or basal diet supplemented with 3% fibrous root processed powder (PR group), in a 42-day experiment. The dietary inclusion of *P. cyrtonema* fibrous roots increased slaughter performance (*p* < 0.05), reduced the fat rate (*p* < 0.05), improved intestinal morphology (*p* < 0.05), and improved the immune organ index to varying degrees. It also significantly improved pH reduction, drip loss, and pressure loss of breast muscle and leg muscle (*p* < 0.05). Furthermore, it significantly improved immune and antioxidant functions including decreased MDA content of serum (*p* < 0.01), increased GSH-Px content (*p* < 0.01), IgG, IgA, and C4 contents (*p* < 0.05), and increased expression of IL-2 and IFN-γ (*p* < 0.01). Additionally, the mechanism by which fibrous roots improve immune function in broilers was explored using network pharmacology and molecular docking technology. Network pharmacology and molecular docking revealed that flavonoids such as baicalein, 4′,5-Dihydroxyflavone, 5,7-dihydroxy-6,8-dimethyl-3-(4′-hydroxybenzyl)-chroman-4-one, and 5,7-dihydroxy-3-(2′-hydroxy-4′-methoxybenzyl)-6,8-dimethyl-chroman-4-one were key components that enhanced immune function through the MAPK1 and other key targets involved in regulating the MAPK signaling pathway. From the findings, it can be concluded that incorporating *P. cyrtonema* Hua fibrous root as a natural feed supplement and growth promoter in broiler diets had a positive impact on bird health and performance.

## 1. Introduction

Antibiotics have played a major role in promoting growth, preventing animal diseases, and improving feed efficiency in poultry production for decades. However, the use of antibiotics used in feed might lead to unfavorable effects on human health [1], pose a great threat to food safety, and negatively impact the ecological environment. Since the 21st century, numerous regulations prohibiting the use of antibiotics in feed have been promulgated worldwide, prompting the poultry industry to focus more on the search for antibiotic alternatives. The main components of various plants, including polyphenols, polysaccharides, terpenoids, and alkaloids, have shown partial positive effects on the growth of poultry [2]. Plant-derived additives can be used singularly or in combination as raw materials in poultry breeding and production, which can alter feed flavor, enhance absorption, regulate immune function, improve growth performance, improve meat quality, and regulate intestinal microbial [3,4]. Moreover, they have many advantages such as being natural, having a wide range of effects, low residue, and no resistance [5], which has attracted extensive attention from scholars at home and abroad.

*Polygonatum cyrtonema* Hua is a perennial herb with a rhizome that has been traditionally used as a food–medicine homologous material. It contains various chemical components, including polysaccharides, steroidal saponins, flavones, triterpenes, alkaloids, and proteins. Among these components, polysaccharides and steroidal saponins are abundant in content, while homosoflavonoids are characteristic substances. The pharmacological effects of *P. cyrtonema* Hua have been reported, such as improving immunity [6], anti-oxidation [7], lowering blood glucose [8], antibacterial and anti-inflammatory effects [9], and anti-fatigue effects [10]. However, due to its long growth cycle and numerous benefits, the resource of *P. cyrtonema* Hua is obviously insufficient, making it increasingly difficult to meet the growing market demand. 

However, in the process of processing *P. cyrtonema* Hua, fibrous roots are usually discarded as waste, accounting for about 15–20% of the total rhizomes. Therefore, reasonable and scientific ways of developing and utilizing the fibrous root of *P. cyrtonema* Hua would be highly significant. Both the rhizome and fibrous roots come from the underground part of *P. cyrtonema* Hua, which means they have common chemical components and similar pharmacological effects to a large extent, such as immunity improvement [10]. This raised the possibility that the discarded roots could be used as an animal feed additive. 

To investigate the nutritional and functional effects of fibrous roots on white-feathered broilers, a study was conducted to examine their effects on growth performance, slaughter performance, immune organs, meat quality, immune parameters, antioxidant function, intestinal absorption, and cytokines. The aim was to explore the feasibility of fibrous roots as animal feed additives and evaluate their use as a natural antibiotic substitute for poultry breeding programs. Additionally, network pharmacology and molecular docking were used to explain the mechanism for pharmacological effect based on the “drug-target-disease-pathway”.

## 2. Results

### 2.1. Contents of Main Chemical Substances

The contents of the main chemical substances in the fibrous roots of *P. cyrtonema* Hua are shown in Table 1. The fibrous root raw powder contained 1.01% total saponins, 0.06% general flavones, 0.80% total proteins, and 8.39% polysaccharides. In the fibrous root processed powder, the concentrations of these substances were 0.51% total saponins, 0.08% general flavones, 0.95% total proteins, and 1.80% polysaccharides.

### 2.2. Growth Performance

The effect of experimental treatments on the growth performance of broiler chickens is shown in Table 2, on days 1 to 21. Compared with the CON birds, the final weights and ADGs of the MD and HD groups significantly decreased (*p* < 0.01), while those in the LD and PR groups were also reduced (*p* < 0.05). Additionally, the F/Gs in the MD and HD group increased significantly (*p* < 0.01 and *p* < 0.05). When compared to the ANT chickens, the final weight and ADG of the MD group reduced (*p* < 0.05). There was no significant interaction of other growth performance parameters between the CON or the ANT group and the experimental groups. It is noteworthy that the antibiotic or the fibrous root of *P. cyrtonema* Hua did not provide any additional benefits to bird growth beyond basal diet, especially during the starter stage. (Appendix A)

From day 22 to day 42, each group had rapid growth in terms of the final weight, ADFI, and ADG. Birds in the LD and PR groups had higher final weights and ADGs when compared to the CON group, although these differences were not statistically significant (*p* > 0.05). In comparison to the ANT group, birds in the MD group had slower increases in the final weight, ADFI, and ADG (*p* < 0.01, *p* < 0.05, and *p* < 0.01, respectively). The LD and PR chickens had better final weights, ADFIs, and ADGs than those in the ANT group, with no significant difference observed. As far as F/G was concerned, there was no significant difference among groups. It was obvious that adding more than 3% of raw fibrous root did not contribute to growth promotion.

Throughout the entire period, the LD broilers had the lowest ADFI, but showed no difference when compared to either the CON group or ANT group. In terms of ADG, both the LD and PR groups demonstrated similar growth-promoting effects as the CON and ANT groups, while the MD and HD chickens had slower growth in ADG. It was observed that the F/G of the LD group was slightly better than those in the CON group and ANT groups, although no difference was found. 

### 2.3. Slaughter Parameters

There were no differences observed in the eviscerated percentage, breast muscle percentage, and thigh muscle percentage among the experimental treatments for broiler chickens, as shown in Table 3. However, a tendency towards an increase was observed when the birds were fed the LD and MD diets. The LD, HD, MD, and PR groups had higher dressing percentage (4.04%, 5.24%, 5.79%, and 5.06%, respectively) than the CON group, with marked increases of 0.321%, 4.21%, 4.74%, and 4.02%, respectively, compared to those of the ANT group. Similar increases were also found when compared to the ANT group. The proportion of abdominal fat in the birds from other groups significantly reduced, with the HD group having the lowest abdominal fat percentage and a decrease of 57.89%. These results indicated that both fibrous root and antibiotics could produce low-fat content and improve muscle gain in broiler chickens.

### 2.4. Meat Quality

For leg muscles, color parameters (L*, a*, b*) showed no difference among all the groups, as depicted in Figure 1, when compared to the CON group. Meats from birds fed with fibrous roots had less pH reduction values than those from the basal diet or ANT diet treatment groups, although this effect was not statistically significant. Drip loss and pressure loss values of muscles in the LD and PR groups had the similar values to those of the CON and ANT groups, except the MD and HD group (*p* < 0.05), in which water-holding capacity was increased. In comparison to the CON group, the dietary supplement with antibiotics and fibrous roots significantly increased the shear force (*p* < 0.05), with the highest increase observed in the HD group at 86.49%. This indicated that fibrous root could make leg muscle tougher.

Similar to leg muscles, color parameters (L*, a*, b*) of breast muscles in all groups were comparable, as shown in Figure 2, especially for L*. After the birds were fed with antibiotics and fibrous roots, a* and b* values of them were higher than those in the basal diet, and the ANT group had numerically improved the b* by 48.62% compared to the CON group. For pH reduction, the addition of antibiotics and root powder resulted in meats having more pH reduction than that of the basal diet. The MD and HD groups had lower drip loss and pressure loss values than those of the CON and ANT groups, and unlike the LD and PR groups, they exhibited better water-holding capacity. Although there were no significant differences in all groups, compared to the basal diet, dietary supplements of antibiotics and root powder led to tougher breast muscles due to a stronger shear force.

### 2.5. Immune Organ Indexes

There were no significant differences in the weights of the liver or spleen among the experimental groups and the CON or ANT group at either 21 days of age or 42 days of age, as shown in Figure 3. The weights of the bursa and thymus were more significantly affected in the birds raised to 42 days of age than 21 days of age. During the starter stage (from 1 to 21 days), the addition of antibiotics or fibrous roots had almost no effect on the weights of the bursa and thymus. However, during the grower and finisher phase (from 22 to 42 days), more increases were observed on the weights of the thymus in the broilers fed with antibiotics or fibrous roots compared to the CON group. In particular, the LD and MD groups had significant increases in thymus weight (*p* < 0.01 and *p* < 0.05, respectively). Additionally, the bursa index of the HD group was higher compared to the CON group (*p* < 0.05) or the ANT group (*p* < 0.05).

### 2.6. Immunoglobulin

The effects of experimental treatments on the immunoglobulins of broiler chickens are shown in Figure 4. On days 1 to 21, compared to the CON group, IgA levels in the LD and PR birds were increased significantly (*p* < 0.01), with the highest increase of 12.99% observed in the PR group. In contrast, the HG group had a marked decreased in IgA compared to the CON group (*p* < 0.01). The experimental groups had no differences from the ANT group except the HG group (*p* < 0.01) and the MD group (*p* < 0.01), which had lower IgA levels. Due to their lower IgG levels (*p* < 0.01) compared to the basal diets, antibiotics and fibrous roots did not seem to improve IgG levels in the young broilers, with only the HD group showing higher IgG levels (*p* < 0.01) than the ANT group. The MD and HD groups increased IgM levels by 22.69% and 12.80%, respectively, over the CON group, while other experimental groups showed no differences. Compared to the ANT group, there was a similar effect on IgM levels. For C3, there was no better improvement in any of the experimental groups, including the ANT group, than in the CON group, and a significant decrease of 12.57% was observed in the LD group (*p* < 0.01). In terms of C4, compared to the CON group, the ANT group and other experimental groups showed better improvement (*p* < 0.01, respectively), except the HD group, which had similar effects.

On days 22 to 42, compared to the CON and ANT groups, fibrous root groups did not show the better effect on enhancing IgA, C3, and C4 in broilers. However, compared to the basal diet and the antibiotic groups, the addition of fibrous roots effectively improved IgG and IgM to a certain extent, with a significant increase in IgG levels (*p* < 0.01).

### 2.7. Effects on Cytokines

On days 1 to 21, as shown in Figure 5, the expression levels of IL-2 and IFN-γ in fibrous root groups, whether raw or processed, were significantly lower (*p* < 0.01 or *p* < 0.05) than those in the basal diet and antibiotics groups. From 22 to 42 days, the expression levels of IL-2 and IFN-γ in the MD, HD, and PR groups showed remarkable enhancement. The MD group had the highest level and was significantly higher (*p* < 0.01) than the CON and ANT groups.

### 2.8. Antioxidative Function

After 21 days of feeding young birds with basal diets mixed with fibrous roots, as shown in Figure 6, the MDA levels in serum were the most remarkably impacted among all the determined antioxidative parameters, including T-AOC, SOD, MDA, and GSH-Px. The MDA levels in the experimental groups were much lower (*p* < 0.01) than those in the CON group and ANT groups. The SOD levels were slightly higher than those in the CON group without any significant difference, but non-significantly lower than those in the ANT group. The GSH-Px levels in the HD group were 20.34% and 16.47% higher than those in the CON and ANT groups, respectively. The T-AOC levels in the MD, HD, and PR groups had significant differences compared to those in the CON and ANT groups (*p* < 0.01), with decreases of 19.44%, 23.61%, and 20.83%, respectively, over the ANT group.

During the days from 22 to 42, there was no significant difference in T-AOC levels between the fibrous root groups and the CON group (*p* > 0.05). However, the MD group had the lowest T-AOC levels with a 17.20% decrease. There was also no difference in SOD levels among all the groups. The MDA levels in the LD, MD, HD, and PR groups had higher increases (*p* < 0.01) than those in the CON group, whereas they were significantly lower (*p* < 0.01) than those in the ANT group. Compared to the CON group, GSH-Px levels in the ANT, LD, HD, and PR groups increased by 30.45%, 18.81%, 28.11%, and 31.25%, respectively. However, there were no significant effects between antibiotics and fibrous root treatments despite a marked decrease of 34.53% in the MD group.

### 2.9. Intestinal Morphology

Figure 7 illustrates the morphometric measurements for the villus height, crypt depth, and villus height to crypt depth (V/C) ratio of broiler chickens. As a whole, the fibrous root groups exhibited an increasing response compared to either the ANT group or CON group. During the days from 1 to 21, when compared to the CON and ANT groups, the MD group had the greatest villus height (*p* < 0.05), which increased by 36.74% and 37.86%, respectively. The LD group increased the crypt depth (*p* < 0.05), and the V/C ratio of the MD group was the highest, being 30.02% higher than that of the CON group and 31.24% higher than that of the ANT group.

During days from 22 to 42, compared to the CON group, the villus heights of the MD and HD groups were 28.42% and 31.21% higher, respectively, while the crypt depth of the HD group increased by 20.42% (*p* < 0.05). No significant difference in the V/C ratio was observed among all the treatment groups. There was no significant difference between the ANT group and any fibrous root treatment group for the villus height, crypt depth, and V/C ratio (*p* > 0.05).

### 2.10. Correlation Analysis

As depicted in Figure 8, there was a positive correlation between total flavonoids and IgG (*p* < 0.001) and IgM (0.01), while a negative correlation was observed with IgA (*p* < 0.001). The total saponins was positively correlated with IgG (*p* < 0.001), IFN-γ, and IgM (*p* < 0.05), but negatively correlated with IgA (*p* < 0.01). The total saponins was positively correlated with IgG (*p* < 0.01) and IFN-γ (*p* < 0.05).

### 2.11. Collection of Flavonoid Targets Related to Immunoglobulin

The PubMed database and TCMSP database combined with the eight flavonoids previously isolated and identified (Table 4) were utilized to mine the potential targets related to immunoglobulins, including IgA, IgG, and IgM. A total of 186 targets were found, and network relationships between these flavonoids and targets were constructed and analyzed using Cytoscape 3.8.0 software (Figure 9). The degrees of these flavonoids were sorted in descending order, as shown in Table 5. Baicalein, Isoflavone 3, DFV, 4′,5-Dihydroxyflavone, and (2R)-7-hydroxy-2-(4-hydroxyphenyl)-chroman-4-one, with a degree of 100, were found to be the most connected targets.

### 2.12. Con3.11 Construction of Protein Interaction (PPI) Network

A total of 533 targets related to IgA, IgG, and IgM were explored through the GeneCards Human database and OMIM database. The intersections of flavonoid and IgA, IgG, and IgM targets were performed, resulting in 17 potential intersecting targets as shown in a Venn diagram (Figure 10). These potential targets of flavonoid-regulated IgA, IgG, and IgM were imported into the STRING database for protein–protein interaction (PPI) network analysis (Figure 11). The interactions between the targets consisted of 17 nodes, 35 edges, 14 correlation targets, and an average degree value of 4.12. The PPI results were imported into Cytoscape 3.8.0 for further analysis, and the flavonoid–immunoglobulin (IgA, IgG, and IgM) target network was generated (Figure 12). The larger the points and the darker the node color, the more significant the target was in the network. The results of various analyses are shown in Table 6. After evaluating the degrees, betweenness, centralities, and closeness centralities of the targets, nine core targets were screened as Mapk3, Stat3, Egfr, Ar, Hdac6, Plg, Mapk1, Insr, and Hdac1.

### 2.13. GO and KEGG Analysis

GO enrichment analysis was performed to obtain GO entries (FDR < 0.01, *p* < 0.01), including BP entries 58, CC entries 9, and MF entries 33. The top 10 entries with the highest number of genes were visualized in Figure 13, revealing the targets involved in the positive regulation of transcription, DNA-templated, peptidyl-tyrosine autophosphorylation, phosphorylation, and insulin-like growth factor receptor signaling pathway, and so forth. These biological processes were regulated by receptor binding, enzyme binding, protein binding and other mechanisms in the caveola, macromolecular complex, axon, and other cellular components.

KEGG enrichment analysis showed 95 pathways (FDR < 0.01, *p* < 0.01) related to the regulation of IgA, IgG, and IgM by flavonoids. The top 10 pathways with the highest number of genes were HIF-1 signaling pathway, FoxO signaling pathway, Chemical carcinogenesis–receptor activation, Adherents junction and others, as listed in Table 7 and visualized in Figure 14. Among these pathways, those directly related to IgA, IgG, and IgM included the MAPK1 signaling pathway, STAT3 signaling pathway, INSR signaling pathway, EGFR signaling pathway, and MAPK3 signaling pathway. The core target-component-pathway network contained 40 nodes and 91 edges with 8 active flavonoids, 9 core targets, and 20 critical pathways, as shown in Figure 15. The regulation of IgA, IgG, and IgM by flavonoids could be characterized by multi-target, multi-compound, and multi-pathway interactions.

### 2.14. Molecular Docking

Among the core targets, there were four potential targets with high degree values, and they were HDAC1(PDB ID:4ZQA), INSR(PDB ID:4VEQ), MAPK1(PDB ID:4S2Z), and STAT3(PDB ID:4ZIA), among which HDAC1 could not bind to small molecular compounds, thus MAPK1, ISNR, and STAT3 were molecularly docked and investigated for possible interactions with 4′,5-dihydroxyflavone, baicalein, isoflavone 1 and isoflavone 3, the top four flavonoids in the target-component-pathway network. The binding energy scores during docking indicated the affinity of a flavonoid for the target protein, less than −5.0 kcal/mol indicated a good molecular docking, and the lower the binding energy score is, the more stable the binding conformation is. Here, the binding energy scores except baicalein-MAPK1 in Table 8 were less than −5.0 kcal/mol, the affinity interactions between MAPK1 and flavonoids were the best; Figure 16 shows the visualization results.

## 3. Discussion

Fibrous root in this research is from *P. cyrtonema* Hua, a food–medicine homologous plant that is widely used as traditional Chinese medicine. It is considered safe enough to be added into broiler diets and can show promising effects in promoting growth. 

The experimental results indicated that both raw and processed fibrous roots could be utilized as adding fodder for white-feathered broilers in terms of growth performance. ADGs, ADFIs, and F/Gs decreased from days 1 to 21 after young chickens were fed diets with fibrous root and antibiotics. Diets with fibrous root contained high amounts of fiber, polysaccharides and saponins, which might cause dyspepsia and loss of appetite in young chickens. On days 22 to 42, as the digestive functions of broilers were improved with enhanced adaptability, ADFIs and ADGs increased rapidly, and F/Gs decreased without any significant difference between the LD group and the CON or ANT group. Overall, adding 1% fibrous root at the grower and finisher stage (days 22 to 42) led to better gains for white-feathered broilers. This suggested that fibrous root had the potential to partially replace antibiotic in promote growth.

Slaughter performance is an important aspect in evaluating the economic performance of meat production [11]. When the dressing percentage is above 85% and the eviscerated rate reaches 60%, it indicates healthy growth and optimal eating performance in broilers [12]. In this study, the mean dressing percentage and semi-eviscerated percentage were found to be 94.11% and 73.50%, respectively. These results showed that the fibrous root of *P. cyrtonema* Hua positively impacted the healthy growth and meat performance of broilers. There was no significant difference in eviscerated percentage, breast muscle percentage, and leg muscle percentage between any fibrous root group and the basal diet group. However, when compared to the ANT group, these parameters showed an increasing trend. Adding fibrous root effectively reduced the abdominal fat rate in the birds, similar to antibiotics, compared to the basal diet. The increase in muscles and decrease in abdominal fat rate indicated that the addition of fibrous root could improve slaughter performance. 

Meat quality assessment is vital for evaluating the physical and chemical properties of broiler meat. Common indicators are meat color (L*, a*, b*), pH value, shear force, drip loss, and pressure loss and so on, which are related to the water-holding capacity, flavor, freshness, and juiciness of meat [13]. Meat color plays an important role in evaluating the appearance of meat and can directly reflect its freshness and quality. L* value is influenced by myoglobin and fat deposition, while a* value is affected by myoglobin and hemoglobin, and b* value is affected by feed coloring [14]. In this experiment, the addition of the fibrous root had little effect on meat color. After slaughtering animals, as storage time increased, lactic acid content in muscle increased continuously, and ATP released H^+^ after hydrolysis. If the pH value drops below the appropriate range, meat quality deteriorates, affecting its flavor [15]. In this experiment, the pH values of chickens decreased after slaughtering but remained within the normal range of 5.4~7.2 [16]. The addition of fibrous root helped delay the decrease in the pH value of leg muscles and maintain stability, but it had the opposite effect on breast muscles. The water-holding capacity and drip loss not only affect its taste and texture, but also impact the storage and transportation of meat products, and a lower drip loss rate and higher water-holding capacity in muscle result in juicier and fresher meat. In this study, drip loss and pressure loss in the MD and HD groups were generally lower than those in the CON group and ANT groups. Shear force typically represents tenderness in chickens, with a negative correlation between shear force values and tenderness. The shear force of the experimental groups increased significantly, leading to relatively tougher meat with lower tenderness. In a word, taking into account all the above indicators, the best improvement in broiler meat quality was achieved by supplementing raw fibrous root at a rate of 3–4%.

The thymus, spleen, and bursa indexes of broiler chickens reflect the growth and development of immune organs, which are the places where immune cells occur, differentiate, and mature. These indicators indicate the strength of the immune function in broiler chickens [17], while liver is mainly responsible for the metabolism and detoxification of broilers. When the bursa and spleen of broilers increase reasonably relative to the weight of broilers, the immune performance is stronger [18]. The test results showed that adding fibrous root could help promote the broilers’ immune organs compared to the CON group and ANT groups, especially the thymus and bursa index, which showed that adding the fibrous root to the basal diet was able to provide enough nutrients for the growth of immune organs and enhance the body immunity.

Immunoglobulin is a class of proteins with antibody activities in serum, playing a vital role in the adaptive immune system of animals. It is widely distributed in the blood, tissue fluid, and fluid, and is divided into IgA, IgG, and IgM based on the differences of heavy chain in their molecular structures [19]. After activation by antigen, antibody complex or microbes with immunological activity, complement also becomes an important part of the immune response [20]. The contents of immunoglobulin and complement directly affect the immunity of poultry. Dietary roots could significantly increase the amounts of IgA and C4 from day 1 to day 21, compared to a basal diet, while also enhancing IgG and IgM contents from day 22 to day 42, compared to antibiotics. The proportion of fibrous root in diet has varying effects on the different indexes; taking the 3% addition as an example, contents of IgG in the HD group had the highest increase, unlike C4 with the lowest value. These results indicate that the fibrous root had specific effects on enhancing the immune performance, reducing the disease occurrence probability, and promoting the healthy growth of broilers.

As a heterogeneous regulator produced by immune cells or non-immune cells in vivo, cytokines play a crucial role in resistance to foreign pathogens by mediating and regulating the immune response and inflammatory response [21]. IL-2 is one of the most widely involved cytokines in the immune response of poultry, promoting the proliferation and differentiation of T cells, generating cytokines and antibodies. IFN-γ can enhance the phagocytic function of macrophages and activate T cells to interfere with virus infection [22]. In this study, the expression of IL-2 and IFN-γ was significantly decreased in broilers fed with the fibrous root of *P. cyrtonema* Hua on days 1 to 21, which seemed to be unfavorable to their immune ability. However, IL-2 and IFN-γ levels were significantly increased on days from 22 to 42 compared to the CON and ANT groups. The results showed that fibrous root could improve the non-specific immunity and cellular immunity function by promoting the cytokine secretion of broilers, particularly for broilers on days 22 to 42. Dietary *Glycyrrhiza* polysaccharide has been considered to significantly improve the expression of IL-1β and IFN-γ in the liver of broilers, and enhance the immune performance of broilers [23].

Overmuch free radicals can result in a negative effect of an oxidative stress response, even the destruction of cell structure by causing cell aging and apoptosis through the peroxidation of the cell biological membrane [24]. In poultry, disease, aging, and other issues caused by the damage of physiological function will directly affect the production performance. Chinese medicinal herbs or their extracts, such as *Polygonati Rhizoma* and Pueraria root, can improve the activities of SOD, CAT, and GSH-Px due to strong antioxidant capacity [25]. In this study, feeding broilers fibrous root significantly reduced the content of MDA in the starter stage while decreasing T-AOC activity. Additionally, it significantly increased the activity of GSH-Px in the grower and finisher stage, suggesting that fibrous root of *P. cyrtonema* Hua had better antioxidant capacity, especially with a 3% addition. 

Intestinal digestion is crucial for nutrient absorption in broilers, and it can be better understood through VH, CD, and V/C. Longer villi increase the contact area between intestinal tract and chyme, thus improving digestive efficiency [26]. Lower crypt depth is better for gut secretion, nutrient absorption, and the regeneration of villous epithelial cells [27]. In this current research, adding fibrous root improved the completeness and neatness of poultry intestinal villi. VH and CD were significantly improved in the LD, MD, HD, and PR groups. It is noteworthy that the MD and HD mixed diets could significantly lengthen intestinal villi, while the PR diet had significantly lower CD with better V/C. This suggested that supplementing had a positive effect on jejunal digestion in broilers. Until now, polysaccharides in Chinese medicinal herbs have been proven to play an important role in improving intestinal structure, nutrient digestion, and utilization in broilers, and promoting their healthy growth [28].

The appropriate dosage of fibrous root in broiler diets is crucial for optimal utilization and feeding efficiency. The indicators mentioned above demonstrated various changes, including decreases or increases. It is essential to comprehensively evaluate these parameters, such as weight gain, growth performance, health status, sustainable development, economic value, and other comprehensive factors. Based on this evaluation, the optimal dose of fibrous raw root of *P. cyrtonema* Hua for broilers was determined to be 4%.

Among these plant substances, such as total saponins, total flavonoids, and polysaccharides, with a strong correlation to indicators of immune function, including IgG, IgM, IgA, C3, and C4, total flavonoids attracted significant attention. Flavonoids were found to regulate IgA, IgG, and IgM through nine core targets, namely, MAPK1, STAT3, Egfr, Ar, Hdac6, Plg, MAPK3, Insr, and Hdac1. MAPK, a class of evolutionarily conserved silk/threonine protein kinases, plays an important role in innate immunity [29]. The transcriptional regulator STAT3 is also critical in vertebrate development and mature tissue function, including the control of inflammation and immunity [30], and plays a central role in regulating the anti-tumor immune response [31].

The results of the GO function enrichment and KEGG pathway enrichment analysis showed that flavonoids mainly act on the MAPK1 signaling pathway, STAT3 signaling pathway, INSR signaling pathway, EGFR signaling pathway, and MAPK3 signaling pathway. Research has demonstrated that MAPK1 can mediate the growth and differentiation of T and B cells; improving the body’s immune capacity by regulating humoral and cellular immunity MAPK3 is essential for inducing T cells and has influence on dendritic cells (DCs, which are professional antigen presenting cells that instruct T cells during the inflammatory course of experimental autoimmune encephalomyelitis), arming T cells in autoimmunity [32]. 

Molecular docking revealed that flavone 2, flavone 4, Isoflavone 1, Isoflavone 3, MAPK1, INSR, and STAT3 had binding energies mostly less than −5 kcal/mol. Notably, the affinity interaction between MAPK1 and flavonoid was the strongest. The key components included baicalein, 4′,5-Dihydroxyflavone, 5,7-dihydroxy-6,8-dimethyl-3-(4′-hydroxybenzyl)-chroman-4-one, and 5,7-dihydroxy-3-(2′-hydroxy-4′-methoxybenzyl)-6,8-dimethyl-chroman-4-one. These compounds affected MAPK1 and other key targets, regulating MAPK signaling pathway and consequently impacting immune function.

## 4. Materials and Methods

### 4.1. Preparation of Fibrous Roots of P. cyrtonema Hua

The raw fibrous roots of *P. cyrtonema* Hua were collected from Yipuyuan Huangjing Technology Co., LTD. in Xinhua County, Hunan Province, China. After drying at 60 °C for 24 h, some of them were milled repeatedly until all passed through an 80-mesh sieve, and the resulting powder was stored at room temperature. The remaining roots were softened completely with water and steamed for 6 h. The steamed juice was reserved for subsequent softening, while the steamed fibrous roots were placed in an oven for air drying at 65 °C for 6 h. This process was repeated once more by softening the steam-dried roots using the reserved steamed juice. After steaming until they tasted sweet, the roots were dried and milled repeatedly until they passed through an 80-mesh sieve. The processed powder was also kept at room temperature.

### 4.2. Determination of the Main Components of Fibrous Roots of P. cyrtonema Hua

The total saponins, general flavones, total proteins, and total polysaccharides contents in the fibrous roots of *P. cyrtonema* Hua were determined using UV spectrophotometry (Spectrophotometer, UV-1900PC, Shanghai Aoyi Instrument Co., Ltd., Shanghai, China). The Vanillin-glacial acetic acid method was used for total saponins, the sodium nitrite method for general flavones, the Coomassie brilliant blue method for total proteins, and the Anthrone-concentrated sulfuric acid method for polysaccharides.

### 4.3. Birds and Diets

The experimental methods were approved by the Institutional Animal Care and Utilization Committee of Hunan Agricultural University; the authorization number is number 2023-101. One-day-old AA-white-feathered broilers (not separated by sex) were purchased from Hunan Shuncheng Industrial Co., LTD.

A total of 360 one-day-old white-feathered broilers (not separated by sex) in good health were individually weighed and randomly allocated into six groups on a weight basis, with each group having 6 replicates and 10 chicks per replicate. The test cycle was only for 42 days and included a 2-phase feeding program: 1 to 21 days of age and 22 to 42 days of age. The control group was fed a basal diet (corn-soybean meal type, CON group), while the experimental groups were fed a basal diet supplemented with antibiotics or different proportions of fibrous root raw powder and processed powder. The antibiotic group (basal diet + 300 mg/kg aureomycin, ANT group), low dose group (basal diet +2% fibrous root raw powder, LD group), medium dose group (basal diet +3% fibrous root raw powder, MD group), high dose group (basal diet +4% fibrous root raw powder, HD group), and processed group (basal diet +3% fibrous root processed powder, PR group). The basal diet was formulated according to the Chicken Feeding Standards (NY/T33-2004), and the nutritional requirements were determined according to NRC1994 broiler amino acid requirements. The dietary ingredients and nutrient levels are shown in Table 9.

Broilers were raised throughout the entire experimental period at the Kaihui Experimental base, Changsha County, College of Animal Science and Technology, Hunan Agricultural University in an environmentally controlled house. The initial temperature was 32 °C, with each weekly temperature drop being 2 to 3 °C until reaching 20 °C; daily light was on for 23 h and off for 1 h. From the first day to the seventh day after arrival, plastic cylinders were used for powder feed and drinking water, with feed given 4 times per day and water given 2 times per day. Granular feed and drinking water were provided using feed tanks and water lines from days 8 to 42, with feed changed once a day. Chicken house cleaning and disinfection were conducted according to routine management procedures.

### 4.4. Data and Sample Collection

#### 4.4.1. Growth Performance

The daily feed intake and body weight (BW) of each cage were recorded. These values were then used to calculate the average daily feed intake (ADFI) and average daily weight gain (ADG) as well as the ratio of feed to weight gain (F/G) for the entire experimental period.

#### 4.4.2. Serum, Immune Organs, and Meat Samples

On days 21 and 42, after a 12-hour fast, six broilers per group were randomly selected for jugular vein blood collection using vacuum vessels. The blood samples were then coagulated at room temperature and centrifuged at 3500 RPM for 15 min. The resulting serum samples were separated and immediately stored at −20 °C for further use. After that, the broilers were euthanized by bleeding through the carotid artery. The thymus, spleen, liver, bursa, and muscle (left pectoral and leg muscles) were then collected for analysis.

#### 4.4.3. Jejunal and Cecum Chyme Samples

At the time of slaughter, a segment of approximately 1 to 2 cm in length from the jejunum of each broiler was removed and separated. The intestinal tissue was then rinsed with 0.9% normal saline and fixed in 4% paraformaldehyde for subsequent analysis of its morphology.

### 4.5. Determination Method

#### 4.5.1. Slaughter Parameters

At 42 days of age, birds were sacrificed after their fasting weights were determined and they were bled through the carotid artery. Hot water was then used to depilate the broilers. The semi-eviscerated birds, fully eviscerated birds, pectoral muscle, leg muscle, and abdominal fat were weighed successively, and the slaughter performance indices were calculated. These specific methods were based on the “Terminology and Measurement and Statistical Methods of Poultry Performance” (NY/T 823-2004, China), which included the following: (1) dressing percentage was the ratio of body weight after exsanguination, removal of feathers, foot cuticle, toe shell, and beak shell to body weight before death; (2) semi-eviscerated percentage was the ratio of the weight after removing trachea, esophagus, cyst, intestine, spleen, pancreas, biliary and reproductive organs, stomach contents, and keratinous membrane on the basis of the slaughter ratio to body weight before death; (3) eviscerated percentage was the ratio of the weight after removing heart, liver, stomach, lung, abdominal fat, head, and feet on the basis of the semi-evisceration ratio to body weight before death; (4) breast muscle percentage was the ratio of the weight of breast muscles to body weight before death; (5) thigh muscle percentage was the ratio of the weight of thigh muscles to body weight before death; (6) abdominal fat percentage was the ratio of the weight of abdominal fat to body weight before death.

#### 4.5.2. Meat Quality

The specific methods were based on the “Agricultural industry standard of the People’s Republic of China” (NY/T 1180-2004, NY/T 2260-2014, and NY/T 2793-2015, China). The color parameters L* (lightness), a* (redness), and b* (yellowness) were measured at 45 min postmortem using a colorimeter (Colorimeter, NR20XE; Shenzhen 3nh Technology Co., Ltd., Shenzhen, China). The pH values (at 45 min and 24 h postmortem) were measured with a pH meter (Testo 205, Teto Instruments International Trading (Shanghai) Co., Ltd., Shanghai, China), and used to calculate pH reduction over 24 h. Water loss was determined using a filter paper press method, while drip loss was scored based on a suspension method, with the weighed leg muscles and breast muscles (2.0 cm × 1.5 cm × 1.5 cm) suspended in the plastic bags and held for 24 h before being reweighed. Shear force was evaluated on meat cores by cutting the meats perpendicularly to the direction of the fiber using a shear instrument (Digital display muscle tenderness meter, C-LM4, College of Engineering, Northeast Agricultural University, Harbin, China).

#### 4.5.3. Antioxidant Activities and Immune Indexes

The antioxidant activities of glutathione peroxidase (GSH-Px) and superoxide dismutase (SOD), as well as the total antioxidant capacity (T-AOC) and malondialdehyde (MDA) levels, were determined in the serum samples on days 21 and 42 using the commercial kits and an enzyme-linked immunosorbent assay (ELISA) kits (Nanjing Jiancheng Bioengineering Institute, Nanjing, China). The immune parameters, including immunoglobulin A (IgA), immunoglobulin G (IgG), immunoglobulin M (IgM), complement 3 (C3), and complement 4 (C4), were also determined using the same kits. The detection process was carried out using a multifunctional enzyme marker (Spark, Hunan Zhike Instrument Equipment Co., Ltd., China) and an incubator (SPX-150BIII, Huanghua Faithful Instrument Co., Ltd., Cangzhou, China). For detailed information on the detection methods, please refer to the website: http://www.njjcbio.com/ (accessed on 20 December 2022).

#### 4.5.4. Histomorphometry of Jejunum

After being removed from 4% paraformaldehyde fixing solution, jejunum samples were embedded in paraffin. Paraffin sections were then prepared and stained with hematoxylin–eosin (HE staining). Five intestinal sections with complete morphology and clear vision were selected, and the villus height (VH), crypt depth (CD), and ratio of villus height to crypt depth (V/C) were measured by microscope image processing software 2.4 (Case Viewer 2.4).

#### 4.5.5. Serum Mucosa Cytokines Expression

The RNA was extracted from the serum samples with the RNA extract kit (Wuhan Xavier Biotechnology Co., Ltd., China). The extracted RNA was then transcribed into cDNA with the Revert Aid First Strand cDNA Synthesis kit (Thermo Fisher Scientific), then RT-qPCR was performed to obtain the *C*_T_ values of interleukin 2 (IL-2) and tumor necrosis factor-γ (IFN-γ) genes for serum cytokines. The relative expression levels of IL-2 and IFN-γ genes were calculated by the 2^−ΔΔCT^ method.

### 4.6. Mechanism for Pharmacological Effect of Fibrous Root of P. cyrtonema Hua on Broilers

#### 4.6.1. Correlation Analysis

Pearson correlation analysis was performed using the Orange pro 2021 software between chemical substances in the fibrous root of *P. cyrtonema* Hua and the above indexes in broilers. The substances and indexes with the highest correlation were selected for further analysis.

#### 4.6.2. Identification of Compounds and Targets

Based on the results of the correlation analysis, flavonoids and immune parameters IgA, IgM, and IgG were mined for further investigation. Eight flavonoids were collected based on the literature and previous research, and targets were retrieved by searching in TCMSP database (https://www.tcmsp-e.com/, accessed on 9 May 2023), STITCH database (http://stitch.embl.de/, accessed on 9 May 2023), and Swiss Target Prediction (http://swisstargetprediction.ch/, accessed on 9 May 2023). The flavonoid–target network map was visualized using Cytoscape (3.8.0).

#### 4.6.3. Screening of IgA, IgM and IgG Related Targets

Using “IgA”, “IgG”, and “IgM” as search terms, their targets were searched in the GeneCards database (https://www.genecards.org/, accessed on 10 May 2023) and OMIM database (https://www.omim.org/, accessed on 10 May 2023). These targets were combined with the previously selected flavonoid targets to obtain the intersection targets using VENNY (2.0.1).

#### 4.6.4. Construction of Protein–Protein Interaction (PPI) Network

The intersection targets were uploaded into the STRING database for PPI network construction. The study species was defined as “Mus musculus”, and the minimum interaction score was set as 0.400, resulting in PPI networks of IgA, IgM, and IgG targets. PPI data were sorted by Cytoscape (3.8.0) to select core target proteins using degree, betweenness centrality, and closeness centrality as reference criteria.

#### 4.6.5. Gene Ontology (GO) and Kyoto Encyclopedia of Genes and Genomes (KEGG) Analysis

The core target proteins were imported into the DAVID database to perform GO and KEGG analysis for biological interpretation. Biological processes (BP), molecular functions (MF), and cell compositions (CC) were displayed by plotting a histogram for GO, and KEGG pathway enrichment analysis was carried out and shown by a bubble chart. The main biological processes and signaling pathways of the interaction between flavonoids and IgA, IgM, and IgG were obtained, and the compound–target–pathway network diagram was made using Cytoscape (3.8.0).

#### 4.6.6. Molecular Docking

The 3D structures of the above flavonoids were downloaded from the PubChem database (https://pubchem.ncbi.nlm.nih.gov/, accessed on 10 May 2023), while the core target proteins were obtained from PCSD PDB database (https://www.rcsb.org/, accessed on 10 May 2023). Autodock software 1.5.6 was used for dewatering and hydrogenation before exporting the structures as pdb files, which were then imported into Autodock tools with the target proteins selected as receptors. After selecting a small molecule as a ligand and setting the twist key, it is exported as a pdbqt format file. The two files were re-imported into Autodock tools using default docking parameters, and the conformation with the lowest binding energy was selected as the optimal conformation after processing. Visualization of optimal conformation was achieved using PYMOL.

### 4.7. Statistical Analysis

Excel 2010 and SPSS 2.0 were employed to process and analyze the data. One-way Analysis of Variance (One-way ANOVA) and Tukey’s Multiple Comparison Method were used to examine the differences between groups. A *p*-value of less than 0.05 was considered significant for all comparisons.

## 5. Conclusions

The inclusion of the fibrous root of *P. cyrtonema* Hua as a natural feed supplement in broiler diets enhanced bird health and performance, such as growth performance, slaughter performance, immune function, antioxidant activity, and intestinal digestion and absorption capacity. During the grower and finisher stages, adding fibrous root showed particularly significant benefits in terms of cytokine secretion and antioxidant activity, leading to improved growth promotion. Moreover, adding the fibrous root of *P. cyrtonema* Hua had similar effects to that of antibiotic supplementation, and in some respects, even surpassed it, suggesting its potential as an alternative to antibiotics as a growth promoter for broiler chickens.

## Figures and Tables

**Figure 1 antibiotics-12-01627-f001:**
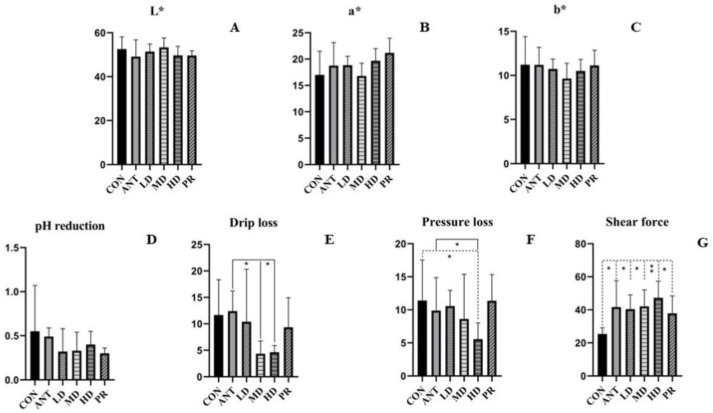
Effects of the fibrous roots of *P. cyrtonema* Hua on leg muscles of white-feathered broilers. (**A**): L*; (**B**): a*; (**C**): b*; (**D**): pH reduction after 24 h; (**E**): drip loss; (**F**): pressure loss; (**G**): shear force. Note: dotted line: compared to the CON group; solid line: compared to the ANT group; * indicates significant differences between groups according to the Tukey test (*: *p* < 0.05; **: *p* < 0.01).

**Figure 2 antibiotics-12-01627-f002:**
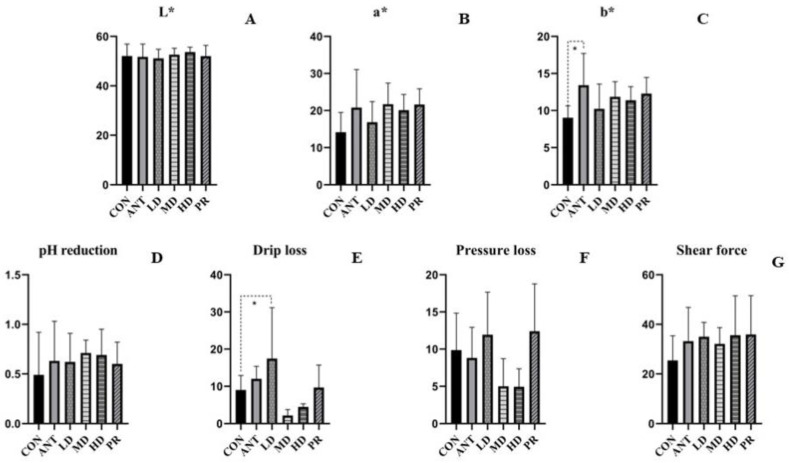
Effects of the fibrous roots of *P. cyrtonema* Hua on breast muscles of white-feathered broilers. (**A**): L*; (**B**): a*; (**C**): b*; (**D**): pH reduction after 24 h; (**E**): drip loss; (**F**): pressure loss; (**G**): shear force. Note: dotted line: compared to the CON group; solid line: compared to the ANT group; * indicates significant differences between groups according to the Tukey test (*: *p* < 0.05).

**Figure 3 antibiotics-12-01627-f003:**
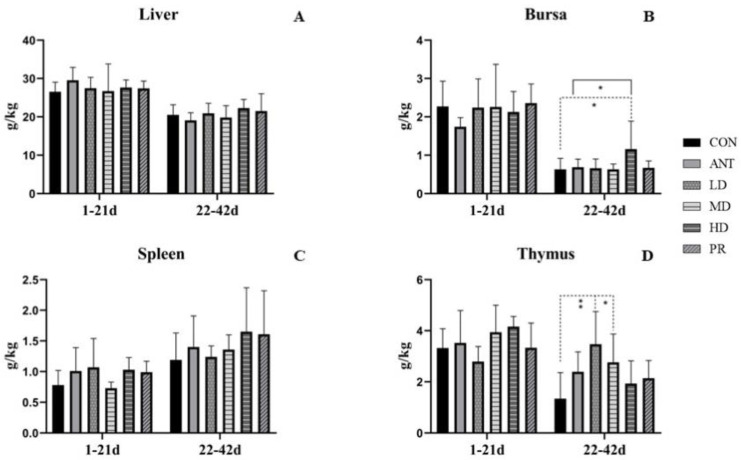
Effects of the fibrous root of *P. cyrtonema* Hua on immune organ indexes of white-feathered broilers. (**A**): Liver organ index; (**B**): bursa organ index; (**C**): spleen organ index; (**D**): thymus organ index. Note: dotted line: compared to the CON group; solid line: compared to the ANT group; * indicates significant differences between groups according to the Tukey test (*: *p* < 0.05; **: *p* < 0.01).

**Figure 4 antibiotics-12-01627-f004:**
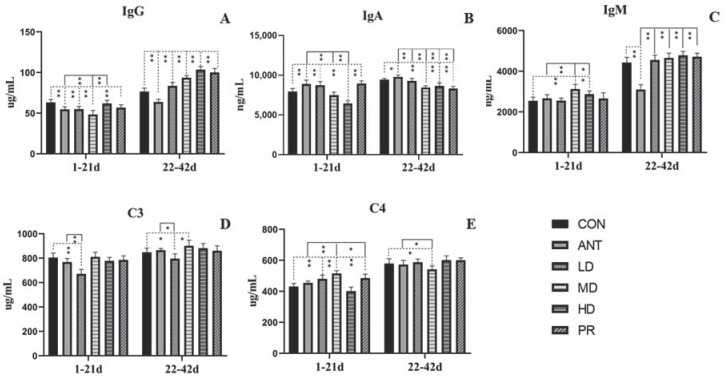
Effects of the fibrous roots of *P. cyrtonema* Hua on immunoglobulin levels in white-feathered broilers. (**A**): Immunoglobulin G (IgG); (**B**): immunoglobulin A (IgA); (**C**): immunoglobulin M (IgM); (**D**): complement 3 (C3); (**E**): complement 4 (C4). Note: dotted line: compared to the CON group; solid line: compared to the ANT group; * indicates significant differences between groups according to the Tukey test (*: *p* < 0.05; **: *p* < 0.01).

**Figure 5 antibiotics-12-01627-f005:**
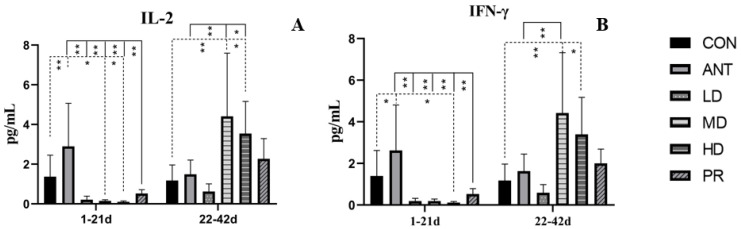
Effects of the fibrous roots of *P. cyrtonema* Hua on mRNA expression levels of cytokines in serum of white-feathered broilers. (**A**): Expression of interleukin-2 (IL-2); (**B**): expression of Interferon alpha-γ (IFN-γ). Note: dotted line: compared to the CON group; solid line: compared to the ANT group; * indicates significant differences between groups according to the Tukey test (*: *p* < 0.05; **: *p* < 0.01).

**Figure 6 antibiotics-12-01627-f006:**
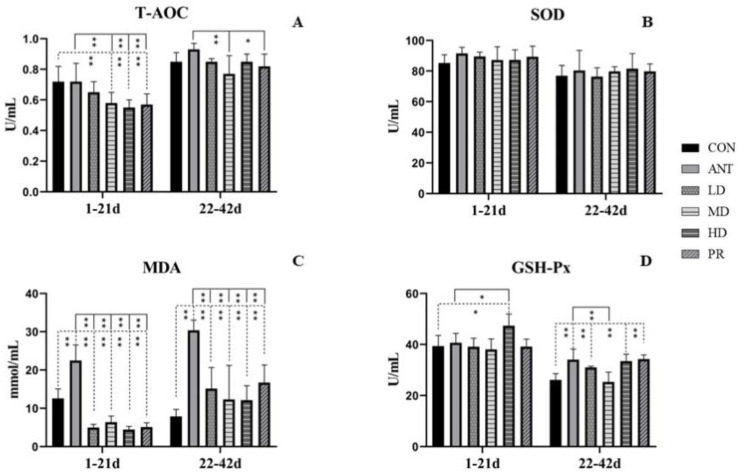
Effects of the fibrous roots of *P. cyrtonema* Hua on antioxidant function of white-feathered broilers. (**A**): Total antioxidant capacity (T-AOC); (**B**): Superoxide dismutase (SOD); (**C**): malonaldehyde (MDA); (**D**): glutathione peroxidase (GSH-Px). Note: dotted line: compared to the CON group; solid line: compared to the ANT group; * indicates significant differences between groups according to the Tukey test (*: *p* < 0.05; **: *p* < 0.01).

**Figure 7 antibiotics-12-01627-f007:**
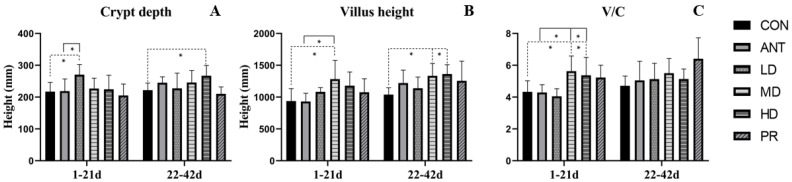
Effects of the fibrous roots of *P. cyrtonema* Hua on intestinal morphology of white-feathered broilers. (**A**): Villus height; (**B**): crypt depth; (**C**): the ratio of villus height to crypt depth. Note: dotted line: compare to the CON group; solid line: compared to the ANT group; * indicates significant differences between groups according to the Tukey test (*: *p* < 0.05).

**Figure 8 antibiotics-12-01627-f008:**
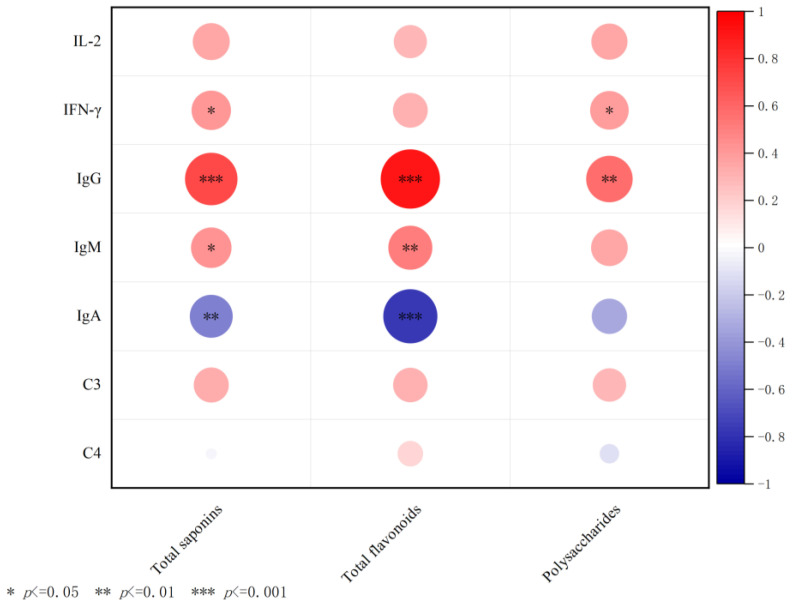
Analysis of the correlation between the main chemical substances in the fibrous roots of *P. cyrtonema* Hua and immune function. Note: * indicates significant differences between groups according to the Tukey test.

**Figure 9 antibiotics-12-01627-f009:**
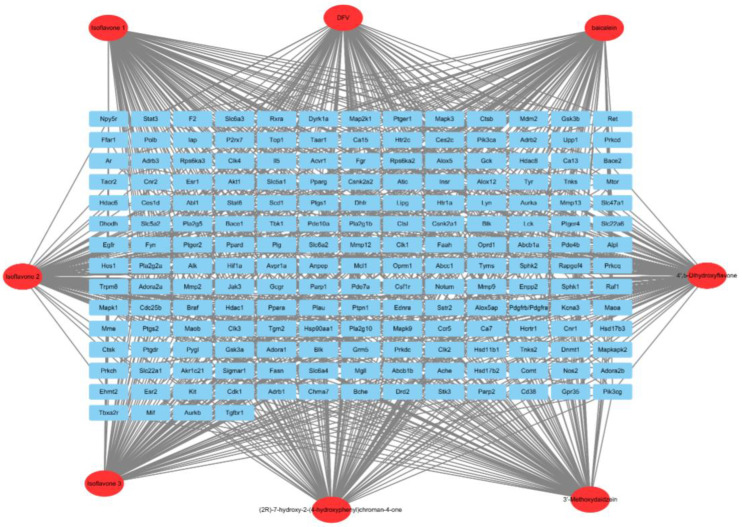
Component-target network of flavonoids in the fibrous roots of *P. cyrtonema* Hua. Note: circular nodes were compounds; rectangular nodes were used as target points.

**Figure 10 antibiotics-12-01627-f010:**
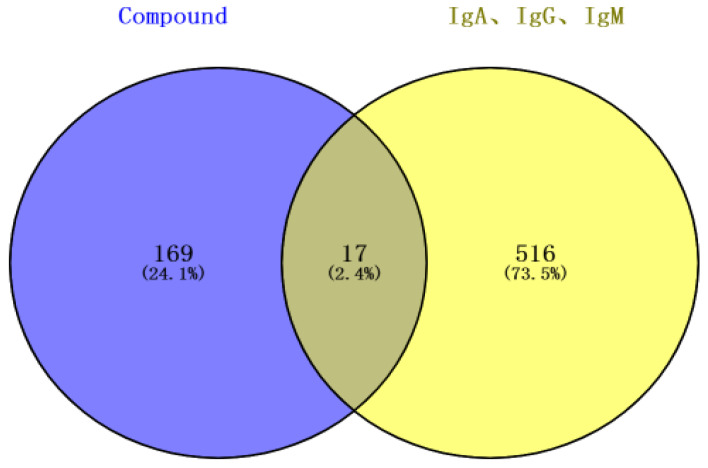
Venn diagram of the intersection between flavonoids in the fibrous root of *P. cyrtonema* Hua and immunoglobulin (IgA, IgG, and IgM) targets.

**Figure 11 antibiotics-12-01627-f011:**
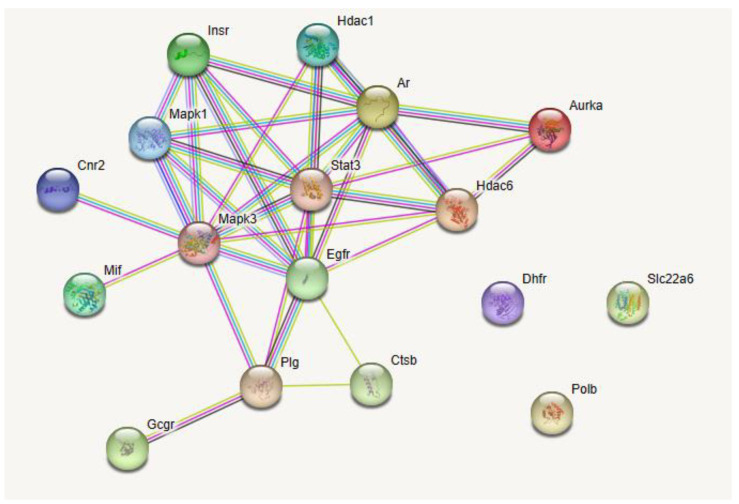
Protein–protein interaction network of the flavonoids in the fibrous root of *P. cyrtonema* Hua and immunoglobulins (IgA, IgG, and IgM) targets.

**Figure 12 antibiotics-12-01627-f012:**
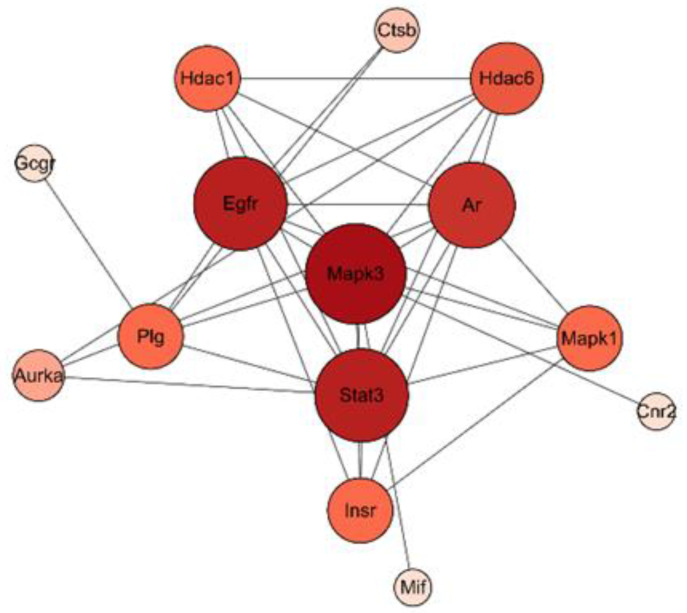
Core target network of the flavonoid–immunoglobulin (IgA, IgG, and IgM) targets.

**Figure 13 antibiotics-12-01627-f013:**
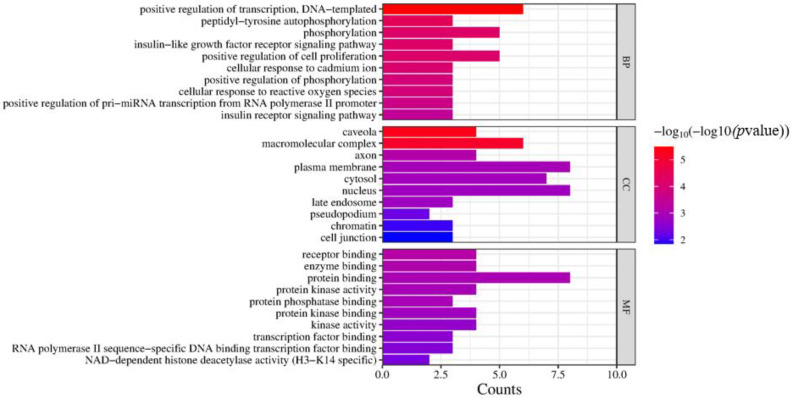
Go enrichment analysis of core targets of the flavonoids in the fibrous root of *P. cyrtonema* Hua for enhancing immune function (IgA, IgG, and IgM).

**Figure 14 antibiotics-12-01627-f014:**
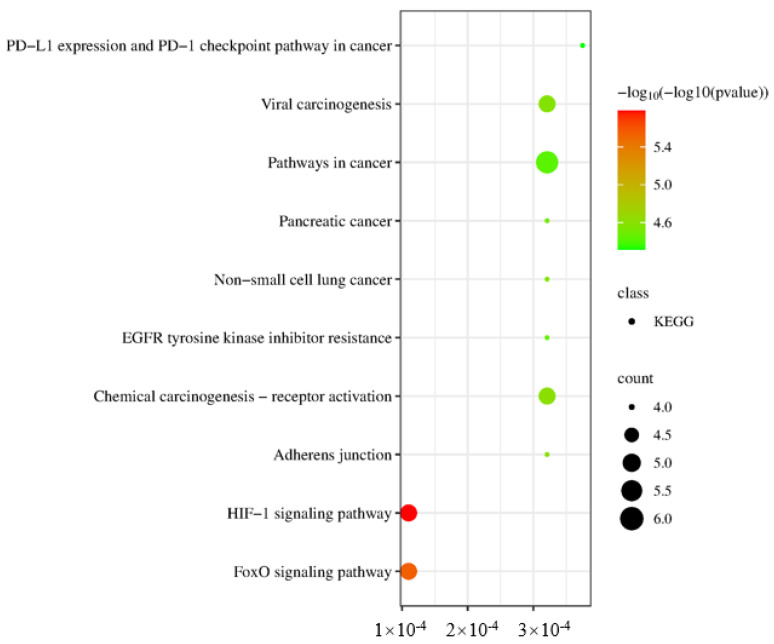
KEGG pathway analysis of core targets of the flavonoid compounds in the fibrous root of *P. cyrtonema* Hua for enhancing immune function (IgA, IgG, and IgM).

**Figure 15 antibiotics-12-01627-f015:**
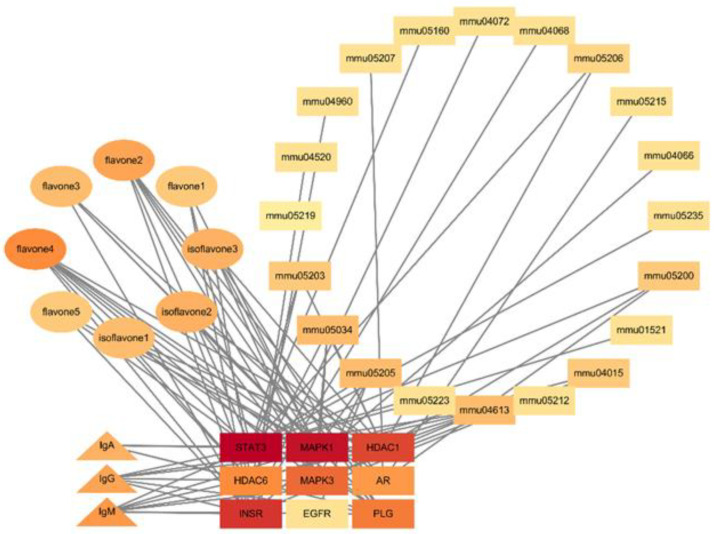
Core target-component-pathway network. Note: ellipse represents compound; triangles represent immune function; rectangles represent pathways and targets.

**Figure 16 antibiotics-12-01627-f016:**
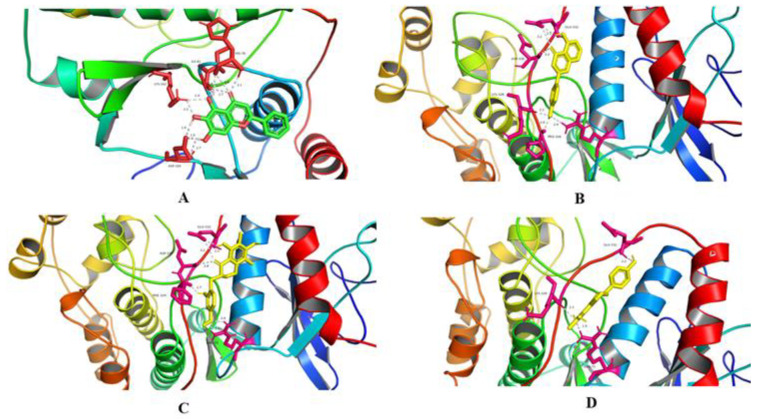
Molecular docking patterns of key components and key targets. (**A**) Flavone 2-MAPK1; (**B**) Flavone 4-MAPK1; (**C**) Isoflavone 3-MAPK1; (**D**) Isoflavone 1-MAPK1.

**Table 1 antibiotics-12-01627-t001:** The contents of main chemical substances in fibrous roots of *P. cyrtonema* Hua (*n* = 3).

No.	Total Saponins (%)	General Flavones (%)	Total Proteins (%)	Polysaccharides (%)
Fibrous root raw powder	1.01 ± 0.05	0.06 ± 0.01	0.80 ± 0.03	8.39 ± 0.19
Fibrous root processed powder	0.51 ± 0.03	0.08 ± 0.02	0.95 ± 0.01	1.80 ± 0.18

**Table 2 antibiotics-12-01627-t002:** Effects of the fibrous root of *P. cyrtonema* Hua on growth performance of white-feathered broilers.

Items	CON Group	ANT Group	LD Group	MD Group	HD Group	PR Group
1~21 d						
Initial weight (g)	46	46	46	46	46	46
Final weight (g)	708.57 ± 34.83 ^Aa^	637.17 ± 99.81 ^ABab^	573.62 ± 95.66 ^ABbc^	511.30 ± 96.62 ^Bc^	552.31 ± 51.58 ^Bbc^	569.91 ± 84.46 ^ABbc^
ADG (g)	31.81 ± 1.85 ^Aa^	27.02 ± 5.56 ^ABab^	24.93 ± 4.67 ^ABbc^	20.78 ± 5.59 ^Bc^	23.64 ± 2.91 ^Bbc^	24.51 ± 4.37 ^ABbc^
ADFI (g)	42.59 ± 2.57 ^a^	41.29 ± 8.39 ^bc^	36.17 ± 5.60 ^bc^	34.14 ± 6.44 ^c^	36.35 ± 4.84 ^bc^	36.80 ± 6.72 ^bc^
F/G	1.34 ± 0.05 ^Bc^	1.54 ± 0.51 ^ABab^	1.47 ± 0.17 ^ABbc^	1.67 ± 0.14 ^Aa^	1.54 ± 0.08 ^ABab^	1.50 ± 0.12 ^ABabc^
22~42 d						
Final weight (g)	2476.67 ± 70.90 ^ABa^	2712.08 ± 321.42 ^Aa^	2559.49 ± 108.56 ^ABab^	2275.04 ± 215.72 ^Bc^	2350.89 ± 158.60 ^Bbc^	2541.91 ± 237.20 ^ABab^
ADG (g)	83.70 ± 3.21 ^Bb^	97.80 ± 12.10 ^Aa^	91.11 ± 4.08 ^ABab^	83.96 ± 6.90 ^Bb^	84.86 ± 6.57 ^ABbc^	92.81 ± 10.26 ^ABab^
ADFI (g)	134.03 ± 11.67 ^ab^	145.68 ± 16.01 ^a^	130.74 ± 16.26 ^ab^	126.56 ± 14.18 ^b^	129.85 ± 12.16 ^ab^	140.00 ± 13.05 ^ab^
F/G	1.60 ± 0.16	1.50 ± 0.15	1.43 ± 0.16	1.51 ± 0.18	1.53 ± 0.16	1.52 ± 0.14
1~42 d						
ADG (g)	63.30 ± 3.66 ^Aa^	63.48 ± 7.65 ^Aa^	59.85 ± 2.58 ^ABab^	53.07 ± 5.14 ^Bc^	54.88 ± 3.78 ^ABbc^	59.43 ± 5.65 ^ABab^
ADFI (g)	88.31 ± 6.86	93.48 ± 9.37	80.60 ± 10.73	88.55 ± 15.47	83.38 ± 5.61	88.84 ± 9.39
F/G	1.39 ± 0.07 ^Bb^	1.48 ± 0.08 ^ABb^	1.35 ± 0.19 ^Bb^	1.67 ± 0.22 ^Aa^	1.52 ± 0.13 ^ABab^	1.50 ± 0.09 ^ABab^

Different letters in the same column indicate significant differences between groups according to the Tukey test; a, b, c and A, B Means in the same row with different superscript letters indicate differences (*p* < 0.05 and *p* < 0.01).

**Table 3 antibiotics-12-01627-t003:** Effects of the fibrous root of *P. cyrtonema* Hua on slaughter performance of white-feathered broilers.

Items	CON Group	ANT Group	LD Group	MD Group	HD Group	PR Group
Dressing percentage	90.57 ± 2.36 ^B^	91.30 ± 1.89 ^B^	94.23 ± 2.05 ^A^	95.14 ± 1.28 ^A^	95.63 ± 1.70 ^A^	94.97 ± 1.32 ^A^
Semi-eviscerated rate	82.19 ± 3.34 ^b^	83.93 ± 3.14 ^ab^	85.09 ± 2.12 ^ab^	86.25 ± 1.52 ^a^	86.25 ± 1.38 ^a^	84.87 ± 2.98 ^ab^
Eviscerated rate	71.61 ± 3.62	72.16 ± 2.80	73.68 ± 2.08	74.28 ± 1.45	74.45 ± 1.04	73.48 ± 2.33
Breast muscle rate	22.18 ± 1.68	21.26 ± 1.34	22.68 ± 1.79	22.72 ± 1.11	22.88 ± 1.09	20.91 ± 2.39
Thigh muscle rate	14.65 ± 0.91 ^ab^	13.59 ± 0.87 ^ab^	14.16 ± 1.83 ^ab^	14.35 ± 0.99 ^ab^	13.41 ± 1.91 ^b^	15.45 ± 1.08 ^a^
Abdominal fat rate	1.52 ± 0.46 ^Aa^	0.74 ± 0.26 ^Bb^	1.01 ± 0.49 ^ABb^	1.03 ± 0.32 ^ABb^	0.64 ± 0.33 ^Bb^	0.91 ± 0.25 ^ABb^

Different letters in the same column indicate significant differences between groups according to the Tukey test; a, b and A, B means in the same row with different superscript letters indicate differences (*p* < 0.05 and *p* < 0.01).

**Table 4 antibiotics-12-01627-t004:** Composition information on flavonoids.

No.	Chemical Name	Type
1	(2R)-7-hydroxy-2-(4-hydroxyphenyl)chroman-4-one	Flavone 1
2	baicalein	Flavone 2
3	3′-Methoxydaidzein	Flavone 3
4	4′,5-Dihydroxyflavone	Flavone 4
5	DFV	Flavone 5
6	5,7-dihydroxy-6,8-dimethyl-3-(4′-hydroxybenzyl)-chroman-4-one	Isoflavone 1
7	5,7-dihydroxy-8-methyl-3-(2′-hydroxy-4′-methoxybenzyl)-chroman-4-one	Isoflavone 2
8	5,7-dihydroxy-3-(2′-hydroxy-4′-methoxybenzyl)-6,8-dimethyl-chroman-4-one	Isoflavone 3

**Table 5 antibiotics-12-01627-t005:** Sorted degrees of flavonoids.

No.	Chemical Name	Degree
1	(2R)-7-hydroxy-2-(4-hydroxyphenyl)chroman-4-one	100
2	baicalein	100
3	5,7-dihydroxy-3-(2′-hydroxy-4′-methoxybenzyl)-6,8-dimethyl-chroman-4-one	100
4	4′,5-Dihydroxyflavone	100
5	DFV	100
6	5,7-dihydroxy-6,8-dimethyl-3-(4′-hydroxybenzyl)-chroman-4-one	99
7	5,7-dihydroxy-8-methyl-3-(2′-hydroxy-4′-methoxybenzyl)-chroman-4-one	97
8	3′-Methoxydaidzein	92

**Table 6 antibiotics-12-01627-t006:** Core targets of flavonoids regulating IgA, IgG, and IgM.

No.	Targets	Degree	Closeness Centrality	Betweenness Centrality
1	Mapk3	10	0.8125	0.35042735
2	Stat3	9	0.76470588	0.12179487
3	Egfr	9	0.76470588	0.15491453
4	Ar	8	0.68421053	0.05021368
5	Hdac6	6	0.61904762	0.02457265
6	Plg	5	0.61904762	0.18269231
7	Mapk1	5	0.59090909	0
8	Insr	5	0.59090909	0
9	Hdac1	5	0.59090909	0
10	Aurka	3	0.48148148	0
11	Ctsb	2	0.48148148	0
12	Mif	1	0.46428571	0
13	Gcgr	1	0.39393939	0
14	Cnr2	1	0.46428571	0

**Table 7 antibiotics-12-01627-t007:** KEGG pathway of flavonoids regulating the enrichment of IgA, IgG, and IgM targets.

No.	Pathways	FDR	*p* Value	Count
1	HIF-1 signaling pathway	1.10 × 10^−4^	1.64 × 10^−6^	5
2	FoxO signaling pathway	1.10 × 10^−4^	2.86 × 10^−6^	5
3	Chemical carcinogenesis–receptor activation	3.21 × 10^−4^	2.45 × 10^−5^	5
4	Adherens junction	3.21 × 10^−4^	2.55 × 10^−5^	4
5	Viral carcinogenesis	3.21 × 10^−4^	2.63 × 10^−5^	5
6	Non-small cell lung cancer	3.21 × 10^−4^	2.66 × 10^−5^	4
7	Pancreatic cancer	3.21 × 10^−4^	3.13 × 10^−5^	4
8	EGFR tyrosine kinase inhibitor resistance	3.21 × 10^−4^	3.52 × 10^−5^	4
9	Pathways in cancer	3.21 × 10^−4^	3.75 × 10^−5^	6
10	PD-L1 expression and PD-1 checkpoint pathway in cancer	3.75 × 10^−4^	4.87 × 10^−5^	4

**Table 8 antibiotics-12-01627-t008:** Binding energy scores of key targets and key components (kcal/mol).

	MAKP1	ISNR	STAT3
Baicalein	−6.94	−4.96	−5.27
4′,5-Dihydroxyflavone	−6.59	−5.39	−5.57
5,7-dihydroxy-6,8-dimethyl-3-(4′-hydroxybenzyl)-chroman-4-one	−6.45	−5.75	−5.77
5,7-dihydroxy-3-(2′-hydroxy-4′-methoxybenzyl)-6,8-dimethyl-chroman-4-one	−6.88	−5.05	−5.23

**Table 9 antibiotics-12-01627-t009:** Experimental basal diet compositions and nutrient levels.

Ingredients (%)	1~21 d (Starter Stage)	22~42 d (Grower and Finisher Stage)
Corn	55.21	62.21
Soybean meal	36.40	29.20
soybean oil	4.70	4.90
Limestone	1.52	1.60
CaHPO4	1.00	1.00
L-Lysine	0.35	0.30
Methionine	0.16	0.13
L-Threonine	0.06	0.06
Salt	0.30	0.30
Premix ^1^	0.30	0.30
Total	100.00	100.00
Nutrient levels ^2^		
Metabolic energy, MJ/kg	12.72	13.02
Curde protein, %	20.65	18.28
Lysine, %	1.27	1.09
Methionine, %	0.47	0.41
Calcium, %	0.90	0.91
Available P, %	0.54	0.52

^1^ Premix is supplied per kg of diet: vitamin A 12000 IU, vitamin D 32500 IU, vitamin E 20.0 mg, vitamin K3 3.0 mg, vitamin B1 3.0 mg, vitamin B2 8.0 mg, vitamin B6 7.0 mg, vitamin B12 0.03 mg, pantothenic acid 20.0 mg, niacin 50.0 mg, biotin 0.1 mg, folic acid 1.5 mg, Fe 45 mg, Cu 17.5 mg, I 1.5 mg, Zn 105 mg, Mn 124 mg, Se 15 mg. ^2^ Nutrient level is calculated value.

## Data Availability

Raw data are held by the authors and may be available upon request.

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
