# Peer review of "Effects of the Fibrous Root of Polygonatum cyrtonema Hua on Growth Performance, Meat Quality, Immunity, Antioxidant Capacity, and Intestinal Morphology of White-Feathered Broilers"

_antibiotics, 2023, doi:10.3390/antibiotics12111627_

Round 1
Reviewer 1 Report
Comments and Suggestions for Authors
Manuscript
Effects of the fibrous root of P. cyrtonema Hua on growth performance, meat quality, immunity, antioxidant and intestinal morphology of white-feathered broilers
This manuscript contains interesting analyses, however there are some important improvements to be done:
Abstract
Suggestion: Avoiding pathogen resistance it’s a research hotspot to find a natural, green, safe and effective plant-derived substitute.
I prefer do not see Drug residues and pathogen resistance in the beginning of the document because this involves some other researches and findings
Please italicize scientific names
L17-18- some suggestions
… cytokines, and antioxidant capacity of white-feathered broilers. Also, the mechanism to improve immune functions of broilers was explored through network pharmacology and molecular docking technology.
L19 – delete with
L21 – replace and the by or a
L22 – delete respectively
L23 – replace can increase by increased
L26 – delete significantly,
L321 – that could
Please include a conclusion in your abstract
Introduction
Please include reference to
.. and also have great damage to the ecological environment [xxxx]
L46, 67, 312… please avoid using like for scientific writing and replace it same comment to “so on” L40, 57….
L62 – replace how to by then
L69 – replace animal food addictive by animal feed additive
L70 – replace investigate by investigated
L72 – replace aimed by aiming
The objective can be clearer in your paragraph, please review it
MM
L83- delete the
Please include a table with the Main Components of fibrous roots of P. cyrtonema Hua as you analyzed
More information is needed on the treatments (fibrous roots characteristics and treat decription)
L96- delete the extra “The”
Please include broilers lineage
The methodology for bird husbandry and diets should be improved
L120 Chicken house cleaning and disinfection was conducted
L86, 117, 704…. Please replace till (not usual for scientific writing) by until
Table 1 – Crude protein
Please include the digestible essential amino acids
Soybean oil
Please include Available phosphorus
Footnotes.. spaces are needed to separate vitamin and name, please review
Since you fed broilers to collect samples, it is important to provide a complete table with feed formulation based on what always we expect
Please be consistent with table format, bold, upper/lower case letters ….
Please use 1 to 21, 22 to 42, or 1 to 42 days (do not separate with - )
Figures are very small and present very low quality, please improve all. They are not helping to see the results
Please present performance and carcass data in Tables (replace Figure 1 by a table, replace Figure 2 by a table) with means, SEM, p-value and mean comparison teste (letters when significant)
Figure 9 can be deleted because nothing is described to demonstrate differences and measurements. Please include this and use this image as supplementary material
Please review and improve conclusions. Also you have to discuss the main points in the discussion, to conclude properly
The last paragraph is not connected with the previous information
Please review all conclusion
Comments on the Quality of English LanguageShould be reviewed and improved
Author Response
Dear Dr.,
Thank you for your decision letter on 10/18/2023 and the reviewers’ comments concerning our manuscript entitled as “Effects of the fibrous root of P. cyrtonema Hua on growth performance, meat quality, immunity, antioxidant and intestinal morphology of white-feathered broilers”. We have read all comments carefully and have made point-by-point responses for every comment. In addition, according to the instructions provided in your letter, we uploaded the files of the revised manuscript. We believe that our revised manuscript is significantly improved and acceptable for publication in Antibiotics.
We would like to thank you for giving us an opportunity to resubmit this revised copy of the manuscript and we really appreciate your time and consideration. We look forward to a positive decision.
Best regards,
Tianlu Zhang
On behalf of all authors
Email: ztl15581233625@163.com

Reviewer 2 Report
Comments and Suggestions for Authors
Effects of the fibrous root of P. cyrtonema Hua on growth performance, meat quality, immunity, antioxidant and intestinal morphology of white-feathered broilers
L16: “Polygonatum cyrtonema” should be in italics.
L13: Change animal breeding to animal nutrition.
L19: Indicate the breed/strain of birds used.
L22-24: Revise “Dietary the fibrous root can increase” to Dietary inclusion of fibrous root of P. cyrtonema increased ….. Also changed all the verbs into past tense.
L28-32: Revise the sentence because it is too long and difficult to understand.
L38-40: This statement is intricate, please revise it: However, several issues of misuse of antibiotics occur frequently such as superposition of excess, timeout and beyond, out of scope, superimposed medication, and so on.
L47-48: Add a reference to the statement.
L50-51: Delete “and so on”.
L54-55: This statement is difficult to understand “Polygonatum cyrtonema Hua is a perennial herb belonging to the genus Polygonatum, as a food-medicine homologous traditional Chinese medicine”.
L63-65: Break the sentence into two.
L70: Change “we investigate” to “the study was conducted to investigate”….
L81: Change “After dried at 60℃” to “After drying at 60℃”… and also indicate the hours the sample was dried.
L82: “80-mesh sieve” What is the actual size of the mesh used?
L83-88: Split the sentence into two or three and also revise them to improve the understanding.
L92: Italicized “P. cyrtonema”.
L96: “The” is repeated, delete one.
L97: AA is first mentioned so write in full.
L99-102: The sentence is too long so break it into two.
L105: Did you mean “aureomycin”?
L111: What do you mean by “NRC1994 broiler amino acid requirements”?
L115-116: Change “with full enclosed chicken coop, automatic facilities control the temperature, humidity and ventilation” to “in environmentally controlled house”.
L116-121: Please revise sentences to past tense.
Table 1: Why was the phosphorus not expressed in “Available P”?
Table 1: Change “early growth stage” to the starter phase and “late growth stage” to the grower and finisher phase.
L129: Change “Then, calculated” to Then used to calculate….
L132: Remove “Growth Performance” before the Serum index.
L133: Change on “On day 21 and 42” to “On days 21 and 42”.
L135: Please revise “After the broilers were killed” to “After that the broilers were euthanized”
L139: Why did you take jejunal samples and not ileal samples?
L146: Change “hot water depilation” to hot water depilated.
The language for the materials and methods, results and discussions as well as the conclusions should be improved. Some of the sentences are very difficult to read and understand.
The presentation of the results needs to be improved. Some of the figures should be converted to tables so that the p-values and the SEM can be shown. Again, some of the figures are too small and that makes reading of the figures difficult.
Some portions of the conclusions look like a repletion of the results and must be revised.
Comments on the Quality of English Language
The quality of the English language used is low and, in most cases, present tenses are used to describe the methodology, results, and discussion.
Author Response

(The authors gave the same response as above.)

Reviewer 3 Report
Comments and Suggestions for Authors
Effects of the fibrous root of P. cyrtonema Hua on growth per formance, meat quality, immunity, antioxidant and intestinal morphology of white-feathered broilers
The paper reported the e effects of different doses of the fibrous root of Polygonatum cyrtonema Hua were evaluated on growth performance, slaughter performance, meat quality, immune function, cytokines, antioxidant capacity of white-feathered broilers. The paper is well written and in line with journal topics. Introduction is clear and complete. Material and methods should be revised, according to the comments reported below and there are some points that should be clarified before publication.
Minor comments
Line 15-18: the sentence is too long: please divide it.
Line 16: change in slaughter parameters
Line 23: delete can
Line 59: change in antioxidant
Line 69 animal feed additive
Line 69-76: the sentence is too long: please divide it.
Line 78: Materials and Methods: please better divide the in vivo sampling, postmortem sampling and lab analyses.
Line 97: please add the number of authorization.
Line 99-101: rewrite for clarity
Line 100: How did the authors choose the sample numerosity? Was the statistical power considered for the investigating parameters?
Line 109: How did the authors choose the dosage of the fibrous root of P. cyrtonema Hua?
Line 140: Add At slaughter.
Line 131: Considering that a positive control with antibiotics was included in the study, did the authors recorded the mortality and morbidity rate in the experimental groups?
Line 132: delete growth performance.
Line 144: change in slaughter parameters
Line 243: Better specify the statistical evaluation, adding the experimental unit. For the blood parameters repeated measure ANOVA is suggested.
Results: to better understand the growth performance and carcass characteristics data, I suggest that the authors include tables instead figures.
Figures: please add the acronyms of experimental diet in the caption.
Line 284: change in slaughter parameters.
Line 315: Usually, the pH value at 45 min and 24 h should be reported.
Line 610: add reference.
Conclusion: the effects of the feed additive is related to the dosage, please consider this in the conclusion section.
Author Response

(The authors gave the same response as above.)
